# Balanced Crystalloids versus Normal Saline in Adults with Sepsis: A Comprehensive Systematic Review and Meta-Analysis

**DOI:** 10.3390/jcm11071971

**Published:** 2022-04-01

**Authors:** Azizullah Beran, Nehaya Altorok, Omar Srour, Saif-Eddin Malhas, Waleed Khokher, Mohammed Mhanna, Hazem Ayesh, Nameer Aladamat, Ziad Abuhelwa, Khaled Srour, Asif Mahmood, Nezam Altorok, Mohammad Taleb, Ragheb Assaly

**Affiliations:** 1Department of Internal Medicine, University of Toledo, Toledo, OH 43606, USA; nehayamunir@gmail.com (N.A.); omar.srour@utoledo.edu (O.S.); saif-eddin.malhas@utoledo.edu (S.-E.M.); waleed.khokher@utoledo.edu (W.K.); mohammed.mhanna@utoledo.edu (M.M.); hazem.ayesh@utoledo.edu (H.A.); ziad.abuhelwa@utoledo.edu (Z.A.); asif.mahmood@utoledo.edu (A.M.); nezam.altorok@utoledo.edu (N.A.); ragheb.assaly@utoledo.edu (R.A.); 2Department of Neurology, University of Toledo, Toledo, OH 43606, USA; nameer.aladamat@utoledo.edu; 3Department of Critical Care Medicine, Henry Ford Health System, Detroit, MI 48202, USA; md-2011@hotmail.com; 4Department of Rheumatology, University of Toledo, Toledo, OH 43606, USA; 5Department of Pulmonary and Critical Care Medicine, University of Toledo, Toledo, OH 43606, USA; mohammad.taleb@utoledo.edu

**Keywords:** normal saline, balanced crystalloids, plasmalyte, lactated ringer, sepsis

## Abstract

The crystalloid fluid of choice in sepsis remains debatable. We aimed to perform a comprehensive meta-analysis to compare the effect of balanced crystalloids (BC) vs. normal saline (NS) in adults with sepsis. A systematic search of PubMed, EMBASE, and Web of Sciences databases through 22 January 2022, was performed for studies that compared BC vs. NS in adults with sepsis. Our outcomes included mortality and acute kidney injury (AKI), need for renal replacement therapy (RRT), and ICU length of stay (LOS). Pooled risk ratio (RR) and mean difference (MD) with the corresponding 95% confidence intervals (CIs) were obtained using a random-effect model. Fifteen studies involving 20,329 patients were included. Overall, BC showed a significant reduction in the overall mortality (RR 0.88, 95% CI 0.81–0.96), 28/30-day mortality (RR 0.87, 95% CI 0.79–0.95), and AKI (RR 0.85, 95% CI 0.77–0.93) but similar 90-day mortality (RR 0.96, 95% CI 0.90–1.03), need for RRT (RR 0.91, 95% CI 0.76–1.08), and ICU LOS (MD −0.25 days, 95% CI −3.44, 2.95), were observed between the two groups. However, subgroup analysis of randomized controlled trials (RCTs) showed no statistically significant differences in overall mortality (RR 0.92, 95% CI 0.82–1.02), AKI (RR 0.71, 95% CI 0.47–1.06), and need for RRT (RR 0.71, 95% CI 0.36–1.41). Our meta-analysis demonstrates that overall BC was associated with reduced mortality and AKI in sepsis compared to NS among patients with sepsis. However, subgroup analysis of RCTs showed no significant differences in both overall mortality and AKI between the groups. There was no significant difference in the need for RRT or ICU LOS between BC and NS. Pending further data, our study supports using BC over NS for fluid resuscitation in adults with sepsis. Further large-scale RCTs are necessary to validate our findings.

## 1. Introduction

Sepsis is defined as life-threatening organ dysfunction caused by a dysregulated host response to infection [1]. Septic shock is defined as a subset of sepsis in which particularly profound circulatory, cellular, and metabolic abnormalities are associated with a greater risk of mortality than with sepsis alone [1]. Despite the declining mortality rates of sepsis over the past decades, the hospitalization for sepsis and its prevalence are increasing [2]. Early resuscitation with intravenous fluids (IVFs) within the first hour (30 mL/kg) is the cornerstone of initial management for severe sepsis and septic shock [3]. Crystalloid fluids are the preferred type of fluids in the resuscitation of septic patients [3]. Crystalloid fluids can be classified to either non-balanced fluids such as 0.9% normal saline (NS) or balanced fluids such as lactated Ringer (LR) or Plasmalyte [4]. Balanced fluids have a more physiologic electrolyte composition closer to plasma compared to NS [4]. NS can induce hyperchloremic metabolic acidosis and has been reported to be associated with acute kidney injury (AKI).

Several meta-analyses have shown lower mortality and AKI with BC compared to NS among critically ill patients, including sepsis and non-sepsis patients [5,6]. However, the crystalloid fluid of choice in sepsis remains debatable. The Surviving Sepsis Campaign International Guidelines in 2016 suggested using either BC or NS for fluid resuscitation of adult patients with sepsis or septic shock [7]. However, the Surviving Sepsis Campaign Guidelines in 2021 suggested a weak recommendation to use BC over NS for fluid resuscitation of adults with sepsis or septic shock [8]. The recommendation was based on low quality of evidence from a network meta-analysis by Rochwerg et al. [9] showed that BC was associated with decreased mortality compared to NS in an indirect comparison. In addition, in the SMART randomized controlled trial (RCT) in 2018 comparing NS to BC, the sepsis cohort was associated with lower 30-day mortality with BC compared to NS (odds ratio (OR) 0.90, 95% CI, 0.67, 0.94) [10]. However, more recent studies have been published comparing the effect of BC vs. NS on clinical outcomes in patients with sepsis and revealed conflicting findings [11,12,13,14].

All previous meta-analyses focused on critically ill patients rather than septic patients [5,6,15]. A recent meta-analysis by Hammond et al. [16] (included 13 trials) found similar 90-day mortality, AKI, and the need for RRT between BC and NS in critically ill patients. This previous meta-analysis conducted a subgroup analysis of the sepsis cohort (included six trials) and revealed no significant difference in 90-day mortality between BC and NS among septic patients. Since lactic acidosis is a significant metabolic side effect of sepsis, patients with sepsis are more susceptible to hyperchloremic metabolic acidosis and AKI compared to other critically ill groups. Therefore, it is more crucial to determine the crystalloid fluid of choice in this selected cohort. To date, no meta-analysis has included all relevant studies and examined all clinical outcomes associated with BC and NS in septic patients solely. There is ongoing uncertainty regarding the choice of crystalloid fluids (BC vs. NS) in the sepsis cohort. As a result, we conducted this comprehensive systematic review and meta-analysis to include all relevant studies to assess the effect of BC vs. NS on different clinical outcomes for adults with sepsis.

## 2. Materials and Methods

### 2.1. Eligibility Criteria

We included studies that met the following eligibility criteria: (1) peer-reviewed cohort studies or randomized controlled trials (RCTs), (2) that compared BC to NS, (3) in patients with sepsis, and (4) reported the outcomes of interest. We excluded conference abstracts. Outcomes of interest included overall mortality at the longest follow-up, 28/30-day mortality, 90-day mortality, AKI, need for RRT, and intensive care unit (ICU) length of stay (LOS). Randomized trials of BC versus NS in critically ill patients were included only if they reported dedicated outcomes in a subgroup of patients with sepsis.

### 2.2. Data Sources and Search Strategy

We performed a systematic search for published studies indexed in PubMed, EMBASE, and Web of Sciences databases from inception to 22 January 2022. We also performed a manual search for additional relevant studies using references of the included articles. The following search terms were used: (“sepsis” or “septic shock”), (“normal saline” or “isotonic saline”), and (“balanced crystalloids” or “lactated ringer” or “plasmalyte”). The search was not limited by language, study design, or country of origin. Appendix A describes the full search term used in each database searched. Two investigators (AB and OS) independently performed the search, screened, and shortlisted the studies for final review. The bibliographic software EndNote was used. Any discrepancies were resolved by a third reviewer (KS). We followed the Preferred Reporting Items for Systematic Reviews and Meta-Analyses (PRISMA) Statement guidelines to select the final studies [17].

### 2.3. Data Extraction

The following data were extracted from the studies: first author name, publication year, country of origin, study design, sample size, gender and age of patients, location of patients, the severity of sepsis, type of BC used, fluid volume, and follow-up duration. Outcome measures were retrieved, including overall mortality at the longest follow-up, 28/30-day mortality, 90-day mortality, AKI, need for RRT, and ICU LOS. Two investigators (AB and OS) independently extracted the data from the included studies. Microsoft Excel was used for data extraction. Any discrepancies were resolved by consensus.

### 2.4. Outcomes of Interest

The primary outcomes of our study were mortality and AKI. The need for RRT and ICU LOS were secondary outcomes.

### 2.5. Statistical Analysis

We performed a meta-analysis of the included studies using Review Manager 5.3 (Cochrane Collaboration, Copenhagen, The Nordic Cochrane Centre) and Comprehensive Meta-Analysis (Biostat, CO, USA). The median and interquartile range were converted to mean and SD where applicable [18]. Given the presumed high heterogeneity in sepsis [19], data were analyzed using a random-effects model and summarized as pooled risk ratio (RR) and mean difference (MD) with the corresponding 95% confidence intervals (CI) for proportional and continuous variables, respectively [20]. A *p*-value <0.05 was considered statistically significant. A fixed-effects model was used alternatively as a sensitivity tool. The heterogeneity was evaluated using the I^2^ statistic as defined by the Cochrane handbook for systematic reviews. I^2^ value of ≥50% was considered significant heterogeneity for all outcomes. We performed subgroup analysis based on the study design (RCTs vs. cohort studies), the type of BC (LR vs. Plasmalyte), and enrollment location (emergency department [ED] vs. ICU) for overall mortality. Sensitivity analysis using leave-one-out meta-analysis was performed, and point estimates were generated.

### 2.6. Bias Assessment

The Jadad composite scale was used to assess the methodological quality of the clinical trials based on randomization, blinding, and withdrawals [21]. The scale ranged from 0 to 5 points 19. Studies with a total score of ≥3 were considered to have a low risk of bias. The Newcastle Ottawa Quality Assessment Scale (NOS) was used to assess the quality of the observational studies based on the selection of the study groups, comparability of study groups, and ascertainment of exposure/outcome [22]. Studies with total scores of ≥6 were considered to have a low risk of bias. For outcomes reported by ≥10 studies, publication bias was assessed qualitatively by visually assessing the funnel plot and quantitively using Egger’s regression analysis. A *p*-value was generated using the Egger analysis, and a value of <0.05 was associated with significant publication bias. Two authors (AB and MM) independently assessed each study for bias. Discrepancies were resolved by a third reviewer (HA).

## 3. Results

### 3.1. Study Selection

Our search strategy retrieved a total of 355 studies. Among these, 36 studies were eligible for systematic review. Subsequently, we excluded 21 studies because of lack of appropriate outcome or population (included non-sepsis patients), secondary paper of included study, or conference abstracts of included or excluded studies. Eventually, 15 studies [11,12,13,14,23,24,25,26,27,28,29,30,31,32,33] met our inclusion criteria and were included in the meta-analysis. Figure 1 shows the PRISMA flow chart that illustrates how the final studies were selected.

### 3.2. Study and Patients’ Characteristics

Table 1 and Table 2 show the study and patient characteristics of the studies included in the meta-analysis. All the included studies were published between November 2013 and January 2022 and included patients with sepsis. Based on country of origin, six studies [24,27,28,29,30,33] originated from the USA, four studies [13,25,26,32] from Asia (China, India, Taiwan, and Thailand), two studies [14,23] from Europe (France and Italy), and two studies [11,31] from Australia and New Zealand, and one study [12] from Brazil. Regarding the design of studies, eight were RCTs [11,12,14,23,25,28,29,31] and seven [13,24,26,27,30,32,33] were retrospective cohort studies. A total of 20,329 patients with sepsis were included. Regarding enrollment location, ten studies [11,12,23,27,28,29,31,32,33] enrolled patients in ICU, four [14,24,25,26] in ED, and one [30] in both ED and ICU. A total of 9752 patients received BC, while 10,577 patients received NS.

### 3.3. Outcomes of Interest

#### 3.3.1. Mortality

Table 2 summarizes the outcomes of the individual studies included in the meta-analysis. Table 3 shows the detailed analysis of all outcomes with subgroup analysis based on the study design (i.e., RCTs and cohort studies).

All 15 studies [11,12,13,14,23,24,25,26,27,28,29,30,31,32,33], which included 20,329 patients with sepsis, reported overall mortality (21.1% in the BC group and 25.2% in the NS group). The overall mortality rate was lower in the BC group compared to NS group (RR 0.88, 95% CI 0.81–0.96, *p* = 0.005, I^2^ = 51%, Figure 2A). The subgroup analysis of observational studies was consistent with lower overall mortality among those received BC compared to NS (RR 0.83, 95% CI 0.71–0.97, *p* = 0.02, I^2^ = 58%, Figure 2B). Subgroup analysis of RCTs revealed a trend toward lower mortality in the BC group but did not reach statistical significance compared with NS (RR 0.92, 95% CI 0.82–1.02, *p* = 0.11, I^2^ = 41%, Figure 2B). Subgroup analysis based on enrollment location (ICU vs. ED) showed consistent results regardless of whether patients were enrolled in the ED (RR 0.73, 95% CI 0.54–0.97, *p* = 0.03, I^2^ = 26%) or the ICU (RR 0.91, 95% CI 0.87–0.96, *p* = 0.0004, I^2^ = 0%), favoring BC over NS in overall mortality (Figure 2C).

No significant difference in overall mortality was observed on subgroup analysis for studies that compared NS with either LR (RR 0.88, 95% CI 0.72–1.07, *p* = 0.21, I^2^ = 43%, Figure 3A) or Plasmalyte (RR 0.97, 95% CI 0.89–1.04, *p* = 0.39, I^2^ = 0%, Figure 3B).

Six studies [13,23,25,28,29,32] reported 28/30-day mortality which showed significantly lower mortality in the BC group compared to NS group (RR 0.87, 95% CI 0.79–0.95, *p* = 0.003, I^2^ = 0%, Figure 3C). The results remained consistent on subgroup analysis of RCTs for 28/30-day mortality (Figure 3C). Five studies [11,12,13,23,31] reported 90-day mortality. However, no statistically significant difference was observed between BC and NS in 90-day mortality (RR 0.96, 95% CI 0.90–1.03, *p* = 0.31, I^2^ = 0%, Figure 3D) with consistent results in subgroup analysis of RCTs.

#### 3.3.2. Acute Kidney Injury

Seven studies [24,25,27,30,31,32,33] (two RCTs and five cohort studies), which included 10,489 patients with sepsis, reported the incidence of AKI (11.3% in the BC group and 12.7% in the NS group). The incidence of AKI was significantly lower in the BC group compared to NS (RR 0.85, 95% CI 0.77–0.93, *p* = 0.0006, I^2^ = 0%, Figure 4A). Subgroup analysis of RCTs showed a trend toward lower incidence of AKI in the BC group but did not reach statistical significance compared with NS (RR 0.71, 95% CI 0.47–1.06, *p* = 0.09, I^2^ = 0%, Figure 4B).

#### 3.3.3. Need for Renal Replacement Therapy

Six studies [14,24,25,27,32,33] (two RCTs and four cohort studies), which included 8358 patients with sepsis, reported the need for RRT. There was no significant difference in the need for RRT between BC and NS groups (RR 0.91, 95% CI 0.76–1.08, *p* = 0.28, I^2^ = 0%, Figure 4C). Subgroup analysis of RCTs showed consistent results (RR 0.71, 95% CI 0.36–1.41, *p* = 0.33, I^2^ = 0%, Figure 4D).

#### 3.3.4. ICU Length of Stay

Three cohort studies [13,32,33], which included 1546 patients with sepsis, reported the ICU LOS. There was no significant difference between BC and NS groups with regard to the ICU LOS (MD −0.25 days, 95% CI −3.44, 2.95, *p* = 0.88, I^2^ = 98%, Figure 4E).

### 3.4. Sensitivity Analysis

Our results remained consistent on the alternative fixed-effects model. A leave-one-out sensitivity analysis for overall mortality, AKI, and the need for RRT revealed consistent results as, shown in Appendix A. Furthermore, the one-study removed sensitivity analysis for AKI showed that excluding the study by Shaw et al. [30] resulted in I^2^ = 11% without significant change in overall mortality, suggesting that the study by Shaw et al. was mainly the reason for the significant heterogeneity in overall mortality (Appendix A).

### 3.5. Quality and Publication Bias Assessment

Quality assessment scores of the RCTs and observational studies are summarized in Appendix A. There was a low risk of bias for all the fifteen studies [11,12,13,14,23,24,25,26,27,28,29,30,31,32,33], as shown in Appendix A. There was a visible asymmetry in the funnel plot of studies that reported overall mortality, which may suggest the presence of publication bias (Appendix A). However, Egger’s regression analysis did not demonstrate a significant publication bias (*p* = 0.19).

## 4. Discussion

In this meta-analysis of 15 studies (eight RCTs and seven cohort studies) that included 20,329 adults with sepsis, BC was associated with lower overall mortality, 28/30-day mortality, and AKI than NS, with similar 90-day mortality, need for RRT, and ICU LOS between BC and NS. However, subgroup analysis of RCTs showed no statistically significant differences in overall mortality, AKI, and need for RRT between BC and NS.

Several meta-analyses have investigated the effect of BC vs. NS among critically ill patients [5,6,15,16,34,35]. Most meta-analyses showed no difference between BC and NS in mortality, the incidence of AKI, and the need for RRT among critically ill patients [5,15,16,35]. Hammond et al. [6] in 2020 demonstrated a lower mortality rate with BC than NS; however, no difference was observed in mortality on the subgroup of RCTs. Recent observational studies and RCTs have been published focusing more on sepsis patients and comparing BC with NS among sepsis [11,12,13,14,25,26]. Some studies have shown mortality benefit favoring BC over NS among patients with sepsis [13,14,27,28]. Although the Surviving Sepsis Campaign Guidelines [8] recently in 2021 preferred BC over NS in sepsis patients, the quality of evidence was low and based on a secondary analysis of SMART trial [10,28].

On the other hand, several recent studies have shown no difference in mortality, the incidence of AKI, and the need for RRT between BC and NS [11,14,24,26]. To our knowledge, no comprehensive meta-analysis in the literature investigated the effect of BC vs. NS among patients with sepsis exclusively and evaluated their impact on different clinical outcomes, including mortality, the incidence of AKI, need for RRT, and length of hospital/ICU stay. Only two meta-analyses by Hammond et al. [6,16] compared BC and NS in critically ill patients and conducted a subgroup analysis of patients with sepsis. One study included only five studies, and the other analysis included six RCTs, and both analyses found no difference in mortality between BC and NS [6,16]. The main limitation of previous meta-analyses was that they focused on critically ill patients with low number of studies that assessed the effect on septic patients. In addition, other outcomes in septic patients, such as incidence of AKI, need for RRT, and length of hospital/ICU stay, were not evaluated in previous meta-analyses. Due to the uncertainty regarding the choice of crystalloid fluids of choice in this cohort of patients, it seems that conducting a systematic review of literature and meta-analysis focusing on patients with sepsis could be beneficial for clinicians and intensivists. Therefore, we provide the first comprehensive meta-analysis to compare BC vs. NS among patients with sepsis and evaluate their effects on different clinical outcomes, including mortality, AKI, need for RRT, and ICU length of stay. Our study included a total of 15 studies for septic patients solely and assessed the effects BC and BC on different clinical outcomes for this selected cohort.

Our overall study results support the recent change in the recommendation from the Surviving Sepsis Campaign Guidelines in 2021, which favored using BC over NS for fluid resuscitation of adults with sepsis or septic shock [8]. Our study demonstrated lower overall mortality and 28/30-day mortality with BC compared to NS. This is consistent with the findings of Raghunathan et al. [27], which showed significantly lower mortality with BC than NS (19.6% vs. 22.8%, respectively) with an RR of 0.86 (95% CI 0.78, 0.94). This difference in mortality between BC and NS is likely attributed to hyperchloremia associated with the use of NS. NS is known to cause hyperchloremia, especially in large volumes, due to its supraphysiologic amount of chloride [27,36]. Studies have shown an independent association between worsening hyperchloremia and mortality [37,38]. In addition, many animal and human studies have shown that hyperchloremia induced from NS leads to renal hypoperfusion and subsequent AKI [39,40]. As a result, AKI is strongly associated with an increased risk of mortality in critically ill patients [41,42]. Interestingly, we observed that the mortality benefit with BC was mainly demonstrated by observational studies [13,27]. Most RCTs demonstrated no difference between BC and NS in mortality among sepsis patients. Only two RCTs [14,28] showed a significant reduction in mortality with BC (*p* = 0.003) among septic patients. The RCT by Young et al. [31] did not show a difference in both mortality and AKI between BC and NS in the sepsis cohort. The mortality and AKI benefit with BC in our study was mainly driven by observational studies rather than RCTs. We believe that these RCTs were likely underpowered to detect the difference in mortality in septic patients. In addition, the lack of significant difference in AKI on subgroup analysis of RCTs was likely due to small number of RCTs (two RCTs). Therefore, more large-scale RCTs are warranted to evaluate the impact of BC vs. NS on mortality and incidence of AKI among septic patients.

Our meta-analysis showed a significant reduction in the incidence of AKI in patients who received BC compared to patients who received NS. This is consistent with the study by Golla et al. [25], which showed a lower incidence of AKI with BC than NS (15.4% vs. 29.1%, respectively, *p* = 0.039). Jaynes et al. [33] also demonstrated a lower AKI with BC than NS (21.3% vs. 30.1%, respectively, *p* = 0.03). The mechanism of AKI induced by NS is not fully understood. The plausible mechanism for NS-induced AKI is the renal hypoperfusion caused by hyperchloremia due to renal afferent arteriole vasoconstriction, especially if given in large amounts [43,44]. On the other hand, a large retrospective study by Zampieri et al. [45] showed that an increase in the amount of LR from 25% to 75% demonstrated a significant reduction in mortality (OR 0.5, 95% CI 0.32–0.79, *p* < 0.001). There was also a linear reduction in AKI incidence as the percentage of LR administered increased (OR 0.99, 95% CI 0.98–0.99, *p* = 0.018) [45]. A further consideration is that, as compared to other critically ill groups, patients with sepsis may be more susceptible to hyperchloremic metabolic acidosis, resulting in greater AKI and, as a result, increased mortality [46]. However, subgroup of RCTs showed no statistically significant difference in AKI between BC and NS. The lack of difference in the subgroup of RCTs is likely due to limited number of RCTs in the analysis. The need for RRT was similar between BC and NS in our meta-analysis for unclear reasons, consistent with previous studies [25,27,32].

Several registered clinical trials are still in the recruitment stage evaluating the effect of BC vs. NS on the clinical outcomes of patients with sepsis, such as RCT by Liu et al. [47] (NCT03685214). These trials are expected to provide more solid evidence regarding the effect of BC and NS among septic patients in adjunct with this meta-analysis.

Several limitations of this study should be acknowledged. First, the meta-analysis included several observational studies with their inherent biases. However, we performed subgroup analysis of RCTs for overall mortality, AKI, and need for RRT, which did not demonstrate significant difference between BC and NS. Therefore, further large-scale RCTs are warranted to validate our findings. Second, the sepsis cohort is a very heterogeneous group to start with, and we could not control for the source and severity of sepsis. Better methods to stratify patients with sepsis are warranted to better understand this heterogeneous population. Third, even though the random-effects model was used in our analysis, there was significant heterogeneity noted in the measurement of some outcomes such as overall mortality and ICU LOS. This might be driven by differences in patient characteristics and sepsis severity and variable follow-up duration across the included studies. Lastly, exposure groups in most studies may not differ sufficiently because most patients did not receive exclusively one crystalloid fluid type. For instance, many studies did not control for fluid administration prior to arrival to ED, such as those given by emergency medical services or fluids received during intravenous medication administration, which might limit our findings.

Despite the limitations, our study has significant strengths. First, we included 15 studies (including nine RCTs) with a total of 20,329 adults with sepsis. To our knowledge, this is the first comprehensive systematic review and meta-analysis combining current manuscripts to compare the effect of BC vs. NS on different clinical outcomes of adults with sepsis. Third, we performed subgroup analysis based on the design of studies (RCTs vs. observational studies) and on the enrollment location (ED vs. ICU) to evaluate the robustness of our results. In addition, our results remained consistent on sensitivity analysis for all outcomes. Lastly, all the included studies were of high quality based on quality assessment.

## 5. Conclusions

Our meta-analysis demonstrates that overall balanced crystalloids were associated with reduced mortality and acute kidney injury in patients with sepsis compared to normal saline. However, subgroup analysis of RCTs showed no significant differences in overall mortality and AKI between the groups. There was no significant difference in the need for renal replacement therapy and ICU length of stay between the groups. Pending further data, our meta-analysis support using balanced crystalloid over normal saline for fluid resuscitation in adults with sepsis. Future large-scale RCTs with better stratification for the source and severity of sepsis are necessary to validate our findings.

## Figures and Tables

**Figure 1 jcm-11-01971-f001:**
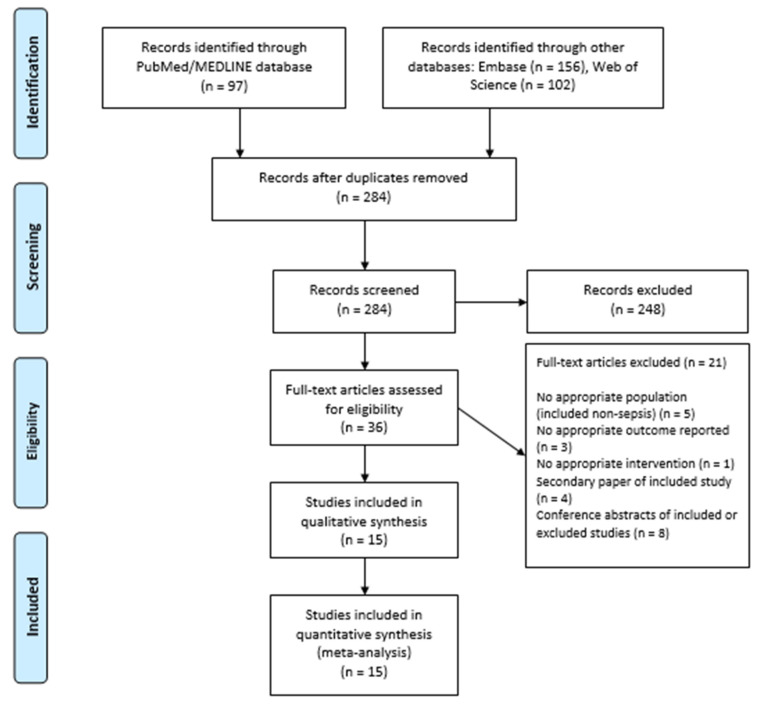
PRISMA flow diagram for the selection of studies.

**Figure 2 jcm-11-01971-f002:**
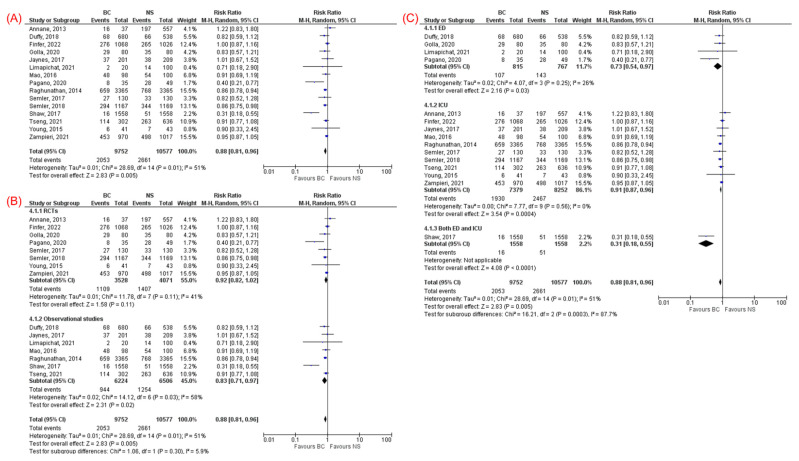
(**A**) Forest plot comparing balanced crystalloids and normal saline regarding overall mortality. (**B**) Subgroup analysis based on the study design (RCTs vs. observational studies) for overall mortality. (**C**) Subgroup analysis based on enrollment location (ED vs. ICU) for overall mortality.

**Figure 3 jcm-11-01971-f003:**
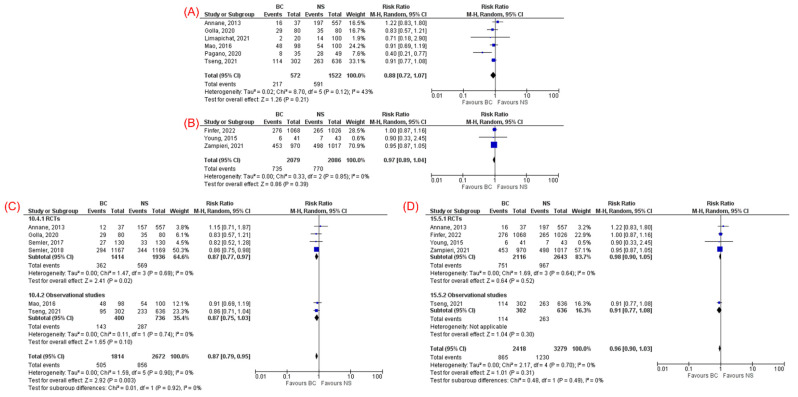
Subgroup analysis based on the type of balanced crystalloids: (**A**) lactated ringer and (**B**) Plasmalyte for overall mortality. Forest plots comparing balanced crystalloids and normal saline regarding: (**C**) 28/30-day mortality and (**D**) 90-day mortality.

**Figure 4 jcm-11-01971-f004:**
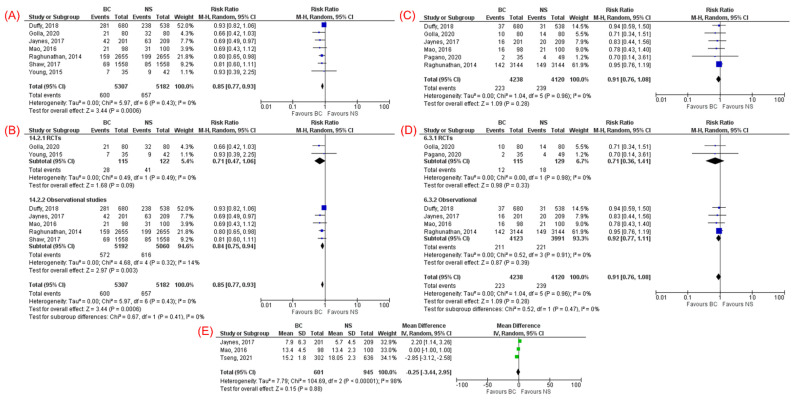
Forest plots comparing balanced crystalloids and normal saline regarding: (**A**) acute kidney injury, (**C**) need for renal replacement therapy, and (**E**) intensive care unit length of stay. Subgroup analysis based on study design (RCTs vs. observational studies) for (**B**) acute kidney injury and (**D**) need for renal replacement therapy.

**Table 1 jcm-11-01971-t001:** Study and patient characteristics of the included studies.

Study, Year	Study Design	Country	Total n (BC/NS)	Male, n	Age, Mean ± SD or Median (IQR), Years	Severity of Sepsis (BC/NS)	Enrollment Location	Type of BC	Fluid Volume (BC/NS), Mean ± SD or Median (IQR) mL	Follow-Up Duration
Annane, 2013	RCT	France	594 (37/557)	NR	NR	NR	ICU	LR	NR	90 days
Duffy, 2018	RC	USA	1218 (680/538)	581	60.6 (18.7)/60.6 (18.7)	qSOFA: 0.68 (0.76)/0.68 (0.76)	ED	Normosol-R	Total: 6000 ± 4600/6500 ± 4800	NR
Finfer, 2022	RCT	Australia and New Zealand	2094 (1068/1026)	NR	NR	NR	ICU	Plasma-Lyte 148	NR	90 days
Golla, 2020	RCT	India	160 (80/80)	85	43.46 ± 17.99/42.44 ± 19.37	SOFA: 7.64 ± 2.56/7.63 ± 2.49	ED	LR	NR	30 days
Jaynes, 2017	RC	USA	410 (201/209)	220	61 ± 14.1/58 ± 14.7	APACHE II: 16.7 ± 6.1/17.3 ± 5.9	ICU	LR and Electrolyte-A	Total: 6750 (4013–10,000)/6500 (4550–12,000)	NR
Limapichat, 2021	RC	Thailand	120 (20/100)	75	69 (59.8–80)/68 (57–82.2)	NEWS: 9 (7, 10.2)/10 (8, 12)	ED	LR	NR	2 days
Mao, 2018	RC	China	198 (98/100)	105	72 ± 9/73 ± 10	NR	ICU	LR	First 72 h: 5092 ± 929/5470 ± 1078	NR
Pagano, 2020	RCT	Italy	84 (35/49)	51	75.9 (12.3)/75.8 (12.1)	SOFA: 5.9 (2.9)/6 (2.8)	ED	LR	First 1 h: 1410/2130	NR
Raghunathan, 2014	RC	USA	6730 (3365/3365)	NR	NR	NR	ICU	NR	NR	2 days
Semler, 2017	RCT	USA	260 (130/130)	NR	NR	NR	ICU	LR or Plasmalyte	NR	30 days
Semler, 2018	RCT	USA	2336 (1167/1169)	NR	NR	NR	ICU	LR or Plasmalyte	NR	30 days
Shaw, 2017	RC	USA	3116 (1558/1558)	1333	NR	NR	ED, ICU and ward	Plasma-Lyte or Normosol	NR	NR
Tseng, 2021	RC	Taiwan	938 (302/636)	707	71.3 ± 15.6	APACHE II: 29 (6.4)/29 (6.4)	ICU	LR	First 24 h: 3172 (2442)/4587 (3776)	90 days
Young, 2015	RCT	Australia and New Zealand	84 (41/43)	NR	NR	APACHE II: 14.1 (6.9)/14.1 (6.9)	ICU	Plasma-Lyte 148	First 24 h: 1200 (0–3000)/1000 (0–3000)	90 days
Zampieri, 2021	RCT	Brazil	1987 (970/1017)	NR	NR	NR	ICU	Plasma-Lyte 148	NR	90 days

Abbreviations: APACHE II: acute physiology and chronic health enquiry, BC: balanced crystalloids, ED: emergency department, n: sample size, ICU: intensive care unit, IQR: interquartile range, LR: lactated ringer, NS: normal saline, NR: not reported, RCT: randomized controlled trials, RC: retrospective cohort, SD: standard deviation, SOFA: sequential organ failure assessment.

**Table 2 jcm-11-01971-t002:** Outcomes of the included studies in the meta-analysis.

Study, Year	Overall Mortality, n (BC/NS)	28/30-Day Mortality, n (BC/NS)	90-Day Mortality, n (BC/NS)	AKI, n (BC/NS)	Need for RRT, n (BC/NS)	ICU LOS, Mean ± SD, Days (BC/NS)
Annane, 2013	16/197	12/157	16/197	NR	NR	NR
Duffy, 2018	68/66	NR	NR	281/238	37/31	4.6/5.6
Finfer, 2022	276/265	NR	276/265	NR	NR	NR
Golla, 2020	29/35	29/35	NR	21/32	10/14	NR
Jaynes, 2017	37/38	NR	NR	42/63	16/20	7 (4–12.5)/5 (3–9)
Limapichat, 2021	2/14	NR	NR	NR	NR	NR
Mao, 2018	48/54	48/54	NR	21/31	16/21	12 (11–17)/13 (12–15)
Pagano, 2020	8/28	NR	NR	NR	2/4	NR
Raghunathan, 2014	659/768	NR	NR	(159/2655)/(199/2655)	(142/3144)/(149/3144)	5.50/5.50
Semler, 2017	27/33	27/33	NR	NR	NR	NR
Semler, 2018	294/344	294/344	NR	NR	NR	NR
Shaw, 2017	16/51	NR	NR	69/85	NR	NR
Tseng, 2021	114/263	95/233	114/263	NR	NR	15.9 (13.7–16.1)/17.8 (16.6–19.7)
Young, 2015	6/7	NR	6/7	(7/35)/(9/42)	NR	NR
Zampieri, 2021	453/498	NR	453/498	NR	NR	NR

Abbreviations: AKI: acute kidney injury, BC: balanced crystalloids, ICU: intensive care unit, LOS: length of stay, NS: normal saline, NR: not reported, n: sample size, RRT: renal replacement therapy, SD: standard deviation.

**Table 3 jcm-11-01971-t003:** Detailed analysis of the outcomes of the meta-analysis with subgroup analysis based on the study design.

Outcomes (Number of Studies)	RR (95% CI)	*p*-Value	I^2^	Subgroup Analysis Based on the Study Design			
				Study Design (Number of Studies)	RR (95% CI)	*p*-Value	I^2^
Overall mortality (15)	0.88 (0.81–0.96)	0.005	51%	RCT (8)	0.92 (0.82–1.02)	0.11	41%
				Cohort (7)	0.83 (0.71–0.97)	0.02	58%
28/30-day mortality (6)	0.87 (0.79, 0.95)	0.003	0%	RCT (4)	0.87 (0.77–0.97)	0.02	0%
				Cohort (2)	N/A	N//A	N/A
90-day mortality (5)	0.96 (0.90–1.03)	0.31	0%	RCT (4)	0.98 (0.90–1.05)	0.52	0%
				Cohort (1)	N/A	N/A	N/A
AKI (7)	0.85 (0.77, 0.93)	0.0006	0%	RCT (2)	0.71 (0.47–1.06)	0.09	0%
				Cohort (5)	0.84 (0.75–0.94)	0.003	14%
Need for RRT (6)	0.91 (0.76, 1.08)	0.28	0%	RCT (2)	0.71 (0.36–1.41)	0.33	0%
				Cohort (4)	0.92 (0.77–1.11)	0.39	0%
ICU LOS (3)	−0.25 (−3.44, 2.95)	0.88	98%	RCTs (0)	N/A	N/A	N/A
				Cohort (3)	−0.25 (−3.44, 2.95)	0.88	98%

Abbreviations: AKI: acute kidney injury, CI: confidence interval, ICU: intensive care unit, LOS: length of stay, N/A: not applicable, RCTs: randomized controlled trials, RRT: renal replacement therapy, RR: risk ratio. N/A: for outcomes that were reported by <2 studies.

## Data Availability

All data underlying this article are available in the article and in its online Appendix A. We will willingly share our knowledge, protocol, and expertise when asked.

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
