# Peer review of "Balanced Crystalloids versus Normal Saline in Adults with Sepsis: A Comprehensive Systematic Review and Meta-Analysis"

_jcm, 2022, doi:10.3390/jcm11071971_

Round 1

Reviewer 1 Report

Thank you very much for the opportunity to review this manuscript and congratulations to the authors for this comprehensive systematic review and meta analysis, which contributes to the important and still controversial topic of fluid resuscitation in critically ill patients. Although the saline versus balanced fluid discussion is not really a new one and the disadvantage of saline in the perioperative setting has been clearly shown, I agree with the authors that the evidence in the cohort of solely septic patients is still weak.

This review and meta analysis is well written, adheres to PRISMA Guidelines and covers all  outcomes of interest.

I only have two comments:

I did not find any description how screening and data collection process was performed: i.e. how many independent reviewers were involved and which tools were used. 

It is a major limitation that this meta analysis has never been preregistered, could you provide any reason for this?

Reviewer 2 Report

The major limitation of current meta-analysis was that similar topic had been assessed in previous meta-analysis. In addition, the inclusion of RCT and retrospective studies as well as different population (e.g., emergency room vs. ICU patients) also make assessment of results difficult. Recent meta-analysis: Zhu Y, Guo N, Song M, Xia F, Wu Y, Wang X, Chen T, Yang Z, Yang S, Zhang Y, Zhang X, Shi Q, Shen X. Balanced crystalloids versus saline in critically ill patients: The PRISMA study of a meta-analysis. Medicine (Baltimore). 2021 Sep 24;100(38):e27203. doi: 10.1097/MD.0000000000027203. PMID: 34559108; PMCID: PMC8462635. Hammond DA, Lam SW, Rech MA, Smith MN, Westrick J, Trivedi AP, Balk RA. Balanced Crystalloids Versus Saline in Critically Ill Adults: A Systematic Review and Meta-analysis. Ann Pharmacother. 2020 Jan;54(1):5-13. doi: 10.1177/1060028019866420. Epub 2019 Jul 31. PMID: 31364382. Abstract: In the abstract, the author reported that subgroup of peer-reviewed studies showed significantly lower AKI with BC (RR 0.85, 95% CI 0.77-0.93). However, the peer-reviewed studied included RCT and retrospective studies. I prefer meta-analysis these results from RCT. Flow chart 1 did not included results from cochrane library. Flow chart 1 demonstrated only 15 studies, but the abstract described that 18 studies were included. Study characteristics: Please described the clinical setting in individual study (e.g, ICU setting or emergency room setting or operation room setting) supplemental figure 1 included retrospective studies?? Figure 2: Did the study by Annane 2018 was RCT? The number of patient in both groups were different. Please confirm this information. Please combined figure 2a and 2b as one figure only. Figure 3. I did not find the information regarding figure E and figure F. Figure 4. I did not find the information regarding figure D

Round 2

Reviewer 2 Report

I thank the authors for addressing my concerns. I had no other questions.